# Understanding climate refugees from educators' perspectives: Social studies teachers' views

**Leyla Donmez Bayrakci**[1]*, **Fatih Ozdemir**[2]

**1** Eskisehir Osmangazi University, Faculty of Education, Eskisehir, Turkey, **2** Yildiz Technical University, Faculty of Education, Turkey

* leyladonmezogu@gmail.com

## Abstract

This study aimed to examine social studies teachers' knowledge, attitudes, and approaches regarding the concept of climate refugees. Employing a phenomenological design within qualitative research, semi-structured interviews were conducted with volunteer social studies teachers from various provinces in Turkey. The data were analyzed through content analysis and rted with direct quotations from participants. Although their knowledge on the subject was limited, social studies teachers demonstrated sensitivity toward climate refugees and made efforts to address these gaps. They attributed the emergence of climate refugees to human activities, particularly political decisions. While they believed that citizenship could be granted to climate refugees under appropriate conditions, they expressed caution due to current refugee-related challenges. The teachers emphasized the necessity of systematically integrating climate refugee issues into the curriculum, raising teachers' awareness, and adopting practical, empathy-based teaching methods.

## Introduction

Migration, a fundamental aspect of human history, is often shaped by environmental factors, including climate change [1]. During the hunter-gatherer periods, people relocated in search of new food sources due to the depletion of local resources and the movement of animals [2].

In later periods, compulsory migrations occurred for various reasons such as the search for pasture, social conflicts, unrest, and invasions. Climate change has historically played a significant role in human mobility, with the migrations in the Tibetan Plateau serving as a clear example [3].

In more recent history, the Migration Period, one of the major turning points in human history, stands out as a striking example of climate-induced migration [4]. This wave of migration led to significant historical consequences, such as the movement of communities, the division of the Roman Empire, and the laying of the foundations for modern European societies [5,4].

**Data availability statement:** All relevant data are within the manuscript and its Supporting Information files.

**Funding:** The author(s) received no specific funding for this work.

**Competing interests:** NO authors have competing interests The authors have declared that no competing interests exist.

Climate-induced migration has been a constant throughout human history and continues to exist today. Africa is the most affected region due to its high climate vulnerability and dependence on nature. Drought lies at the root of rural conflicts and migrations in many different regions [6],Tongwane, Ramotubei, & Moeletsi, 2022, [7]. Groups that have to leave the region they live in due to climate change are called Environmental Migrants. International Organization for Migration [8] defines environmental migrants as "individuals or communities who migrate temporarily or permanently within their countries or across borders due to sudden or gradual environmental changes that adversely affect their lives or living conditions, for compelling reasons or by their own choice." As we get closer to the present day, it is seen that this term has shifted from the definition of environmental migrant to the definition of climate refugee. [9,10,11,12,13,14,15]. A climate refugee is an individual who migrates permanently or temporarily to a different region due to environmental challenges caused by climate change.

Due to its geographical location, Turkey functions as a bridge between Asia, Africa and Europe and has historically been on an important migration route [16–18]. North Africa, the Middle East and Turkey within the Mediterranean Basin are among the regions with high vulnerability to global climate change [19]. In this context, climate change in the Mediterranean Basin strengthens the possibility of an increase in the number of climate refugees from the region in the medium term [20]. According to the report published by the Internal Displacement Monitoring Centre (IDMC) in 2025, 83.4 million people worldwide became internal migrants by the end of 2024. 9.8 million of these people were subjected to climate and disaster-induced displacement [21]. The report states that disaster-induced migration is particularly high in South Asian countries such as India, Pakistan and Bangladesh. Positioned in the climate-vulnerable Mediterranean Basin and on key migration routes from Asia to economically prosperous Europe, Turkey is likely to face increasing pressure from climate-induced migration. Therefore, it must develop effective strategies to manage this emerging challenge.

Although refugees are displaced for different reasons, they face similar problems during and after the migration process. In these processes, problems such as language barriers, economic difficulties, sociocultural incompatibility, traumatic experiences, and social isolation are frequently experienced [22,23]. Different language, religion, and cultural structures encountered after migration can also be an important source of stress for individuals [23,24]. One of the main problems faced by migrants and refugees is education []. Education is a critical issue for climate refugees, who face challenges similar to other migrant groups but often under more severe conditions due to long-term environmental [25,26,27,28–31,32] and economic vulnerabilities. As many climate refugees come from low-income communities, they are likely to encounter unique structural inequalities in education. Although this study does not directly focus on climate refugee education, it explores educators' perspectives—an essential foundation for shaping educational responses.

Various solutions have been proposed to address the challenges in the education of climate-affected and displaced populations. Strategies such as flexible learning,

language and cultural adaptation, psychosocial support, and capacity building have been implemented in the Asia-Pacific region [33,34]. However, these often require structural reforms and significant resources. Given the high number of refugees and economic constraints, there is a need for more practical approaches within existing systems. In this context, social studies courses play an important role in supporting the integration of refugee students [35,36]. Defined by Barr, Barth, and Shermis (1977) [37] as citizenship education based on human relations, social studies aim to equip students with the knowledge, skills, and values necessary for active civic participation [38]. The core components of social studies include knowledge, information processing skills, democratic values, and social participation [39].

Published in 2024, the Social Studies Curriculum supports this aim and aims to "provide individuals with the skills they need in social issues based on changing and developing world conditions" [40,41]. In line with all this information, the importance of the social studies course is increasing in issues that facilitate adaptation to the culture of the country such as adaptation to social life, environmental sensitivity, immigrant education, citizenship education and empathy in changing world conditions. In the effective execution of this process, the knowledge and attitudes of teachers who transfer course content to students about climate refugees play a decisive role.

The literature includes numerous studies on the impact of climate change on migration [42,11,43,44,45], [15], the legal status of climate refugees [9,46,47, 48], and their security challenges [14]. Research also addresses refugee integration into education systems [49,50,51], common issues in refugee education [52,53, 54,55,56,29,27,25], and teachers' perspectives on refugees [57,58]. Some studies also explore how social studies education can support refugee students [35,36]. However, there is no research focusing specifically on social studies teachers' views on climate refugeeism. Considering their role in connecting environmental issues with migration education, social studies teachers are vital in addressing current and future challenges. This study contributes a unique perspective and offers foundational insights for teacher education and training programs.

## Method

### Research design

Since this study aims to examine the perceptions of social studies teachers about climate refugees, it was conducted with phenomenological design, one of the qualitative research designs. According to Husserl, phenomenology is a way of reaching knowledge based on phenomena [59]. Phenomenon refers to our experiences in our daily lives as individuals and the meanings of our experiences [60]. In this study, "climate refugeeism" is considered as a phenomenon. It is aimed to be analyzed in depth based on the experiences of teachers regarding the concept of climate refugeeism. Phenomenological research is the collective state of a particular group's experiences about a concept or phenomenon [61]. In accordance with the purpose of the study, it was conducted with a phenomenological design. In this study, data collection from participants began on **16/05/2024** and was completed on **16/05/2025**.

### Study group of the research

While forming the study group of the research, maximum diversity sampling, one of the purposeful sampling types, was used. In maximum variation sampling, the aim is to identify and define the main themes that include many differences of the subject being researched [60]. Demographic information about the participants of the study is presented in Table 1.

### Data collection tool and data collection

In the study, a semi-structured interview form developed by the researcher was used to reveal the views of social studies teachers on climate refugees. During the semi-structured interview, probe questions are organized to make the participant's views on the subject more detailed, to provide clarification or to enable them to give examples [62]. While preparing the semi-structured interview form in the research, draft questions were prepared by examining the literature. The draft

**Table 1. Distribution of teachers by city, duration of occupation, education level and gender.**

| Participant | City | Department | Year | Education Level | Gender Male/ Female | Interview Duration |
|---|---|---|---|---|---|---|
| K1 | Karaman | Social Studies Teacher Education | 20 | PhD | M | 26 |
| K2 | Istanbul | Social Studies Teacher Education | 2 | License | M | 16 |
| K3 | Malatya | Social Studies Teacher Education | 3 | Master | M | 14 |
| K4 | Konya | Social Studies Teacher Education | 12 | License | F | 17 |
| K5 | Istanbul | History Teaching | 30 | License | M | 31 |
| K6 | Mardin | Social Studies Teacher Education | 2 | License | M | 18 |
| K7 | Istanbul | Social Studies Teacher Education | 6 | License | M | 16 |
| K8 | Ankara | Social Studies Teacher Education | 3 | License | F | 15 |
| K9 | Kocaeli | Social Studies Teacher Education | 7 | License | F | 14 |
| K10 | Istanbul | Social Studies Teacher Education | 12 | PhD | M | 20 |
| K11 | Şırnak | Social Studies Teacher Education | 8 | Master | F | 18 |
| | | | **Total Duration** | | | 204 minutes. |

questions were presented to two field experts and a language expert. With the feedback received from the experts, the form was reorganized and the pilot application was started. In the pilot study, one question was found to be difficult to answer and was removed from the form.

During data collection, brief preliminary interviews were held with volunteer participants to assess their familiarity with the term "climate refugees." Findings showed that most teachers were unfamiliar with the concept. Afterwards, two instructional sessions—including articles, news content, and documentaries—were conducted. Initial interview results are addressed in the findings and discussion sections.

After the online lessons, face-to-face interviews were conducted with teachers in the nearby region, and interviews were conducted with teachers in different provinces through the online system. The interviews were conducted personally by the researcher in a quiet and appropriate environment. The interviews were recorded with the help of a voice recorder with the permission of the participants. The interviews were conducted by the researcher himself and did not involve third parties and lasted an average of 18 minutes.

## Data analysis

Data analysis in qualitative research involves preparing data, coding, creating themes and presenting the obtained data in tables, direct quotations or descriptive texts [61]. In this study, content analysis, one of the qualitative research data analysis methods, was used. Content analysis aims to structure voluminous qualitative data into smaller and more meaningful units in line with recurring words, expressions and themes [60]. The data obtained were transcribed with the Transcriptor software and checked for accuracy by the researcher. The analysis process was initiated by reading or listening to the data repeatedly by the researchers; expressions with similar meanings were coded and grouped, and themes were created in line with these codes [63]. Participants were anonymized in accordance with the purpose of the study and teachers were represented with codes such as "T-1", "T-2"... "T- 12". The data were presented under themes and the findings obtained were supported by direct quotations from the participant opinions and a qualified interpretation process was realized.

## Reliability of qualitative data

In qualitative research, validity and reliability are addressed through the criteria of credibility, transferability, dependability, and confirmability, as proposed by Guba and Lincoln (1982), in contrast to the internal/external validity and reliability

concepts of positivist research [64,65]; Öztürk Fidan & Fidan, 2018]. In this study, various strategies were employed in line with these criteria.

To ensure credibility, explanatory probe questions were used during teacher interviews; data were coded independently by two researchers at different times, inter-coder agreement was ensured, and expert feedback supported the analysis process. The significance of the findings was strengthened by comparing them with the existing literature. For transferability, a maximum variation sampling method was adopted, interviews were conducted with all participants, and detailed participant profiles were provided.

To ensure dependability, the data collection and analysis procedures were described in detail; interviews were audio-recorded in line with ethical standards, and independent coding by two researchers was compared for consistency. Regarding confirmability, the literature was reviewed prior to the study, the interview form was developed with expert input, and pilot interviews were conducted. Interviews were held in appropriate settings, and triangulation was employed throughout the research process to enhance researcher coordination. As noted by Stake (1995)[66], triangulation helps reduce bias and increase research validity. Interpretations were supported by direct quotations, and the findings were discussed in relation to the literature.

## Artificial intelligence statement

Artificial intelligence was utilized during the research process to enhance brainstorming, provide code suggestions, ensure code compliance, and correct spelling and language errors, streamlining development and improving overall quality.

## Findings

In this research, which aims to examine the views of social studies teachers on climate refugees, the main themes of "Knowledge and Experience Regarding the Concept of Climate Refugee, Sharing Responsibility, Future of Climate Refugee, Comparison with Other Refugees and Citizenship, Awareness of the Concept of "Climate Refugee" and Its Handling in the Lesson, Methods and Techniques in Teaching the Concept of Climate Refugee" were reached during the research process.

### Knowledge and experience theme regarding the concept of climate refugee

The theme of knowledge and experience regarding the concept of climate refugee is divided into two sub-themes: "Knowledge of the Climate Refugee Concept" and "Experience with Climate Refugees."

### Knowledge of the climate refugee concept

The theme of Knowledge of the Climate Refugee Concept consists of two codes: Those Who Do Not Know the Concept and Those Who Know the Concept. Participant opinions for these codes are presented separately below:

P3: *"It might refer to a concept related to climate. Since it's called a refugee, it could be someone who migrates forcibly."*

P6: *"We attended a seminar on migration, about refugees, but it wasn't related to climate."*

Upon examining the opinions, it is observed that participants are unaware of the climate refugee concept and interpret it solely based on its name. One teacher, despite attending a migration seminar, stated they did not know the concept; others made only speculative comments.

P4: *"There are trainings on climate refugees in EBA's education sections. There are trainings on climate crises. I've seen these before, in national and international media as well."*

P10: *"I've come across this concept before because I've read some literature, but I haven't received any training on it."*

Upon examining the opinions, it is evident that P4 is aware of the concept through EBA trainings and media, while P10 knows it through academic literature but has not received formal training. Generally, the proportion of those who know the concept is low, and those who do know it have gained this knowledge through personal interest or academic studies.

### Experience with climate refugees

The sub-theme of Experience with Climate Refugees consists of two codes: Those Without Experience and Those With Experience. Teacher opinions for these codes are presented below:

P2: *"No. This is a natural disaster, but a different kind of natural disaster."*

P5: *"We haven't encountered it, nor were we aware of it. I know about it, though. Maybe there are such cases."* P8: *"No, they generally said they came either because of war or for better education."*

It is understood that participants have generally encountered individuals who migrated due to war or other natural disasters. P5, while stating they are unaware of the concept, suggests they might have encountered such individuals.

P6: *"I've heard of people coming from Iraq and Syria. Syria is very current. We see people migrating from Mardin due to drought and insufficient land."*

P7: *"Today, people coming from Syria, Iraq, those southern regions… They come not only because of war but also due to climate reasons."*

P10: *"There's such a situation in the Southeast. Desertification is increasing significantly. For example, there's incredible desertification in Mardin. Additionally, there's a situation affecting Şanlıurfa and Şırnak."*

Upon examining the opinions, it is observed that, due to Turkey's geographical position, teachers have experience with both national and international climate refugees. P6 and P7 emphasize that while people from Syria and Iraq are generally considered war refugees, they also migrate due to climate-related reasons. P10 draws attention to migrations in the Southeastern Anatolia Region caused by drought and desertification.

### Sharing responsibility

The theme of Sharing Responsibility is divided into two sub-themes: "Human" and "Non-Human".
**Human induced.** The sub-themes Human İnduced is divided into two sub-themes: " Politicians" and "All Humanity". Participant opinions for these codes are presented separately below:

P1 *"State policies are the factors that trigger this climate crisis and thus climate refugees"*

P9 *"Here, too, the policies of the state were probably more effective. That's why we can say state administrators or people who are responsible for preventing it or delaying it."*

When the views of the participants are analyzed, it is seen that politicians are the most important factor in the emergence of climate crisis and climate refugees and the delay in the solution of the problems.

P1 *"Plus people's living arrangements, growing conditions, people who grow up with an economical living arrangement, for example, pollute the world less."*

*P3 "Because the general cause of climate changes is caused by humans. For this reason, some people are victimized by them."*

Analysis of teachers' opinions reveals that some teachers think that the main factors contributing to climate change and the emergence of climate refugees are unconscious human behavior and misuse of nature.

**Non-human factors.** Teachers' views on the "Totally Inhuman" and "Partially Inhuman" codes in the Non-Human Factors sub- theme are presented below;

*P2 "Then there is nothing people can do here. It's natural."*

*P1- "But those who migrate based on climate events, that is, mostly natural events force them. So you cannot prevent it after a point."*

When the participant views are examined, P2 emphasized that the main factor in the emergence of climate change and climate refugees is entirely nature, while P1 emphasized the partial effect of nature.

### The future of climate refugees

The Future of Climate Refugees theme is divided into two sub-themes: "Climate Change and International Measures" and "Climate Refugees in Turkey."

**Climate change and international measures.** Teachers' opinions on "The Necessity of International Measures" and "Climate Change and Resource Scarcity" under the sub-theme of Climate Change and International Measures are presented below:

*P5: "Apart from climate change, the United Nations should be prepared to provide emergency water or food aid in such places in a way that does not trigger sudden migration."*

*P10: "But of course, this requires both the empowerment of the public and a more active role for decision-makers."*

Participants emphasized the need to raise public awareness and take measures at the national and international levels against the negative consequences of climate change.

*P1: "In the past, the water supplier in the village served 4-5 people at the same time. Now it is not even enough for one person."*

*P6: "Our village in the Mardin region has been legendary for its water supply for years. Now it is facing drought. While our village had over 100 households, it has now dropped below 50."*

Teachers have directly experienced examples of water and food shortages due to drought caused by climate change and have pointed out that this situation will increase the number of climate refugees.

### Turkey-centered climate refugees

**Climate refugees in Turkey.** Teachers' opinions on the codes "Turkey in a Dangerous Region" and "Turkey is Not in Danger" under the sub-theme of "Climate Refugees in Turkey" are given below:

*P4: "I think Turkey, which is located on migration routes, will be one of the countries most affected by this situation."*

P8: *"Especially from the Konya region... Despite being a very large region, it has a very small population, and we explain in textbooks that it is a region that sends migrants, with decreasing rates."*

Upon examining the opinions, P4 stated that Turkey will be affected by climate refugees due to its location on migration routes. P8 emphasized that the Konya region is taught in classes as an area that sends migrants and that the number of climate refugees may increase.

P2: *"From my perspective, I don't think it is a very disadvantaged climate region."*

In this example, the participating teacher stated that Turkey is not a very disadvantaged region in terms of climate.

**Comparison with other refugees and consideration in the context of citizenship.** This theme is divided into two sub-themes: "Comparison with Other Refugees" and "Consideration in the Context of Citizenship."

**Comparison with other refugees.** The sub-theme Comparison with Other Refugees consists of the codes "Harmless, Irreversible Process, Fewer Victims, Large Scale, No Difference." Separate teacher opinions are presented for each of these codes.

P1: *"But the climate crisis, meaning that climate is above many things and is a rule of nature, so people may think there is not much that can be done and approach these individuals with a bit more leniency."*

P10: *"This man, so to speak, is poor. He has no religion, ethnic identity, or race. Therefore, of course, he is first in line in terms of climate refugee experience."*

According to the participants' views, since human-made discrimination or violence do not play a significant role in the situations climate refugees face, they may be subject to a more lenient approach, as they are perceived to be less exposed to discrimination and more victimized by nature.

P8: *"When the country recovers from war, they can return. But since the climate cannot recover and the situation will continue to worsen, they may not be able to return."*

P11: *"Other refugees can return to their countries when the war ends or other circumstances cease to exist, but since the climate is not something that can be fixed, this is somewhat more of a necessity."*

It was emphasized that other refugees can return to their countries when the reasons for seeking asylum disappear, but climate refugees do not have this opportunity due to the irreversible nature of the climate.

P7: *"For me, refugees fleeing war are the top priority. Because they are losing their lives more quickly and their safety is much more important, there are concerns."*

P9: *"Actually, other refugees are more easily recognized by people because they came due to war or larger events. But the situation is not so clear with climate refugees."*

Participants believe that refugees fleeing due to reasons such as war are more victimized because they are at greater risk of losing their lives. Climate refugees, on the other hand, may be more difficult to identify because their reasons are less clear.

P4: *"War can happen in very small geographical areas, but since the climate crisis will spread over much larger areas, I predict that there will be many more climate refugees than war refugees."*

This statement emphasizes that the climate crisis will affect much larger geographical areas and displace more people.

P5: *"They should be treated the same as those who come from war. They should be given the same status. I don't know what that status is, though."*

This opinion expresses that climate refugees should be treated with the same status as other refugees.

**Addressing citizenship**

The sub-theme "Addressing Citizenship" consists of the codes "Abuse, Citizenship Should Be Granted, Citizenship Should Be Granted, and Dependent on a National Referendum." The participants' views on the codes are presented below;

P2: *"It's as if there are too many legal loopholes... I thought that people with bad intentions could use this as an opportunity."*

P8: *"Are they really coming for that reason? This is important. For Syria, we took in people who said they came from the war, but were they all really fleeing the war?"*

Participants emphasized that some people could abuse this legal right and migrate without a real reason. P8 also noted that it is unclear whether Syrian migrants came because of the war.

P1: *"Because the likelihood of the climate conditions in the region they came from returning to their previous state is very low, perhaps even impossible. So where will that person return to?"*

P11: *"I think it should be granted. Because they cannot use the natural resources in their environment due to the climate."*

In these views, it was stated that climate refugees could be granted citizenship because the conditions in their regions of origin are no longer suitable for human life.

P7*: "Whatever the form of government in that country is, if it is a republic, a democracy, if the people have a say, a referendum can be held. The opinion of the public should be considered; do the people want this or not?"*

This statement indicates that the granting of citizenship should be dependent on the decision of the people of that country.

**Awareness of the concept of climate refugees and its treatment in class**

This theme is divided into two sub-themes: "Whether It Is Taught in Class" and "The Necessity of Teaching It in Class."
**Whether It Is Taught in Class.** The sub-theme of Whether It Is Taught in Class consists of the codes "Indirectly Addressed in Class," "Directly Addressed in Class," and "Not Included in the Curriculum." Teachers' opinions within the scope of the codes are presented below:

P2: *"I taught it, but I didn't know it was called 'climate refugee.' I explained it to the children as natural and economic migration."*

P7: *"I touched on climate issues in this way. Climate is already in the book, to be honest. Global warming, climate... When these topics are asked, we inevitably touch on them, even if we don't use the term directly."*

When the participants' views were examined, it was seen that the teachers did not know the concept of "climate refugee" directly, so they addressed the topic indirectly in line with the meaning of the concept.

> P10: *"Of course. Three years ago, there was a major fire disaster in Australia, and there were migrations from there to Indonesia and even New Zealand. Based on this case study, we hadn't fully established the concept, but what was it? We discussed the activities of people who were forced to leave their country due to natural disasters."*

Upon examining P10's statement, it is evident that the teacher specifically refers to migrations in the East Asia and Pacific region, where climate migration is most clearly defined in the literature and legal processes are underway. This demonstrates the participating teacher's mastery of the subject and the literature.

> P8: *"Actually, even when social studies textbooks mention climate, they are so superficial that it would be very good to have lessons to raise awareness among children and protect nature. We have separate topics on migration and climate. Neither of them is addressed."*

P8 emphasizes that the topics of climate and migration are addressed separately, but the issue of climate refugees is not addressed and the explanation remains quite superficial.

**Necessity of teaching in class.** The sub-theme of Necessity of Teaching in Class consists of the codes "Necessity in the Context of Climate" and "Necessity in the Context of Society." Teachers' opinions regarding the codes are presented below:

> P1: *"It should definitely be taught. It is already being taught, albeit to a limited extent. The concept of refugeeism is not covered much, but global warming and the consequences of other natural disasters, migration events, etc. are already being taught."*

> P10: *"This should definitely be covered in social studies classes, and this concept should even be mentioned in the textbook. It should be included in the curriculum. When discussing global climate change, we would be remiss not to mention its impact on refugees."*

P1's statement emphasizes that the topics are generally covered, but the name of the concept is not mentioned. P10, on the other hand, argues that this concept should be explicitly included in the curriculum and textbooks.

> P3: *"Because social studies is a course where sciences related to society are taught and how they can be used in students' daily lives is discussed."*

> P11: *"So, if the place of climate refugees in daily life and how they can be integrated into life is discussed in class and students become aware, maybe they will do something for the climate. For the future."*

Participants emphasized that, by its very nature, social studies teaches topics related to daily life and therefore long-term issues that directly concern society, such as climate refugees, should be included in the curriculum.

## Methods and techniques in teaching the concept of climate refugee

The theme of methods and techniques in the education of the concept of climate refugee is divided into six sub- themes: "Activity-Based Methods", "Learning through Experience", "Collaboration with Society and Institutions", "Teacher Training and Awareness Raising", "Media Use" and "Empathizing".

**Activity-based methods.** The activity-based methods sub-theme consists of four codes: "Brainstorming", "Presentation and projects", "Case study method" and "Theater and drama". Teacher opinions on these codes are presented separately;

P2 *"Since it is a new concept, brainstorming can be done. I think that a short information can be given and then the main information can be presented, such as what the children think about this subject first.*

When the statement is examined, while emphasizing that the brainstorming technique will be effective in teaching new concepts, *P11 "Brainstorming technique can be used after watching something."* In the related statement, he emphasized that the brainstorming technique would be effective in teaching the concept of climate refugee after watching something on the subject.

P3 *"Presentations about climate change can be prepared. The events that cause climate change can be addressed and this situation can be tried to be prevented"* emphasizes that the presentation will be effective, *while P11* states that students will learn more effectively with projects in the example of *"And the importance of the climate issue, what should be done to prevent climate victimization, maybe with the invention method...the project method can be done."*

Participant 1 *"I use the case study technique here the most"* and P9 *"I think it would be more effective to reach a conclusion, an inference through case studies."* emphasize that the case study technique will be effective in concept teaching.

Participant 7 said, "*I mean, for example, there are awareness theaters. The children who are processed on this can see more concrete*." He states that the theater method will be effective in terms of concretization in learning the concept of climate refugee.

In the sub-theme of activity-based methods, teachers emphasized that brainstorming, presentations and projects, case studies and theaters would be effective in teaching the concept of climate refugeeism.

**Learning through experience.** The sub-theme of learning through experience consists of three codes: "Out-of-school learning environments", "Learning by doing and experiencing" and "Sustainable life awareness". Teacher opinions on these codes are presented separately

As an example of out-of-school learning environments, *P10 said, "Especially once it makes sense here to take students' out-of-school learning to the fields. Why? Because of the following, brother, now I think taking these children to a forest area and then showing them desert environments through a virtual museum provides a creative comparison here. Because they will say, "I can make an economic wealth comparison between the desert and places with more agricultural land or forested areas."* and *P11 "I mean, there can be excursions. Or what can happen? How we can use climate-related resources, how we can turn them into energy."* It can be given. Participant statements emphasize that out-of-school learning environments, one of the basic learning methods of environmental education, will be effective in teaching the concept of climate refugeeism.

As an example of learning by doing and experiencing technique, *P8 said "If children learn by doing and experiencing, if they realize something, if they are confronted a little bit, I think they can leave traces of life.* " and *P6 "If the child can resist for 3-5 days in a dry area or if the child does not carry the water in the empty field with a bottle, flask, jerry can and irrigate something and does not do this for days and months and does not see that difficulty, it can show the value of water."* can be given. The participant emphasizes that students should face climate change and drought through learning by doing.

As an example within the framework of sustainable life awareness, *P1 said, "As I just mentioned, how can I sustain my life with the least damage to nature in every step they take in daily life? They will be concerned about this. How can I live*

*economically? They will be concerned about this".* P1 emphasized that students should be aware of sustainable living in order not to negatively affect climate change.

**Cooperation with society and institutions.** Cooperation with Society and Institutions sub-theme consists of two codes: "NGO cooperation" and "Academic support/ University cooperation". Teacher opinions on these codes are presented separately as an example of the NGO cooperation code, P1 "*The work of non-governmental organizations related to this climate crisis throughout the country is brought to the agenda, and these are evaluated in the lessons." and P10 "According to the organization such as TEMA, Kızılay, you will send personnel directly to the school so that children are in direct contact with them. You have to involve the school in them and them in the school.*" P1 and P10 emphasize that non-governmental organizations related to the environment should actively participate in schools and lessons.

As an example of academic support and university cooperation *P5 said "What can be done? If there is a study about this at your university, visuals, publicity can be made, teachers can be invited. These can be done on the basis of consciousness and volunteerism among people like you who volunteer, work and think about doing this job, such as a seminar.*" emphasizes that academics who conduct academic studies should conduct awareness- raising studies for schools and increase academic studies.

**Teacher training and awareness raising.** There is a "Seminar" code in the sub-theme of Teacher Training and Awareness Raising. P8 and P5 statements related to this code can be given as examples.

*P8 "Conferences can be organized. Teachers can also raise awareness among children. If we were a little bit conscious, for example, I never knew about it." P5 "Awareness among seminars, these can be done on a voluntary basis. It can even be done in schools. Certain pilot schools are selected. It also raises awareness with teachers.*"

When the statements of the participants are examined, it is emphasized that seminars and trainings should be given mostly for teachers in order to raise awareness in schools.

**Media use.** The sub-theme of media use includes the code "Visual material and media use". The statement of P7 regarding this code can be given as an example; *P7 "Since we are in the digital age, inevitably television, press, internet should be used. We reach the widest audience with these. Either these issues can be covered on television with state support, such as public service announcements."* When the statement of participant seven is analyzed, all inventories of the media are an effective method in teaching the climate refugee topic.

**Empathizing.** The code "Refugee narratives, testimonies" is included in the Empathizing sub-theme. The statements of P5 and P6 regarding this code can be given as examples;

*P6 "Or exposing people to this situation, these refugees, these climate refugees, migrating from here and coming and talking to them, living in that society, that is, needing to live. It would be good if they can stay."*

*P5 "There are studies about those who have suffered from it, not the video. Surveys, interviews, interviews with those people, and what kind of climatic change and destruction it has caused in those countries, how and what people have experienced, should be conveyed in a way that will remain in the memory, even if briefly."*

When the participant statements are examined, it is emphasized that empathy and learning will be easier if the experiences of climate refugee individuals are conveyed to students first-hand.

In this section, which is based on the findings from the participant interviews, it was concluded that most teachers were unfamiliar with the concept of climate refugees. On the other hand, it was observed that some participants had experiences with climate refugees in their daily lives. Social studies teachers stated that politicians and humanity as a whole bear the greatest responsibility for the emergence of climate refugees, while two teachers emphasized non-human factors. Participants drew attention to the need for international measures regarding the future of the concept of climate

refugees; one teacher stated that Turkey is at risk, while another stated that Turkey will not be affected by this situation. Compared to other refugees, participants seem to have adopted a more moderate approach towards climate refugees. In the context of citizenship, some teachers thought that this situation could be abused, while others took a moderate approach to granting citizenship. One teacher argued that citizenship should be determined by a public vote. In social studies classes, one teacher stated that the concept of climate refugee was addressed indirectly, while another stated that it was not included in the curriculum. Teachers emphasized that this concept must be included in social studies classes, highlighting climate and social reasons. In addition, emphasis was placed on the use of active learning techniques in teaching the concept of climate refugee.

## Conclusion and discussion

This study aimed to examine social studies teachers' views on climate refugeeism. Addressed within the context of social change driven by global climate change, climate refugeeism is considered a key learning area in social studies, particularly at ages when students begin developing awareness. Teachers' knowledge and attitudes, given their central role in instruction, directly influence educational quality. This research seeks to address a significant gap in the literature, with findings discussed considering existing theoretical and empirical work.

Based on participant interviews, most teachers were unfamiliar with the concept of climate refugeeism, while those who were informed had gained this knowledge through personal interest, media, or academic sources. This may stem from the concept's lack of legal recognition [67,46,9,Yücel, 2020, 68,48]. Some participants, however, reported real-life encounters with individuals fitting the definition of climate refugees. These findings reflect conceptual ambiguity, as the term evolved from "environmental migrant" [69] to "climate refugee" in subsequent literature [9,10,11,12,13,14,15]. This evolution may explain why teachers lack formal knowledge of the term but still possess relevant experiences.

In addition, the participants' experiences of climate change and climate refugees in the country can be supported by the literature. According to the IPCC (2022) [19] report, Turkey is among the regions with high vulnerability to global climate change. This situation brings drought and regional vulnerability [70,71]. In this context, climate change in the Mediterranean Basin strengthens the possibility of an increase in the number of climate refugees from this region in the medium term [20]. Some of the participants emphasized that climate may also be among the reasons for migration of migrants of Iraqi and Syrian origin. This result is in line with the studies of Meier, Bond & Bond (2007) [6] and Tongwane, Ramotubei & Moeletsi (2022) [72], who found that climate is the main reason for African origin migration mobility.

Social studies teachers largely hold politicians responsible for the emergence of climate refugees, aligning with literature that highlights the role of human-induced policies in climate change [,19,73,74]. Participants also attributed responsibility to all of humanity. Similarly, IPCC (2021) [75] identifies human activity as the primary driver of climate change, while Öner (2023) [76] emphasizes anthropogenic factors alongside natural processes. Docherty & Giannini (2009) [46] frame climate change and its consequences as both moral and legal responsibilities of humankind. Other studies [77,78,79,80,81,82] underscore the need for individual and collective responsibility for environmental protection. These findings support the view that environmental degradation and climate-related displacement are primarily human-induced. However, two participants attributed such changes to non-human causes, reflecting a possible psychological tendency to deny human accountability [83].

In the study, the participants emphasized that international measures should be taken regarding the future of the concept of climate refugeeism. In their study, Black et al. (2011) [84] stated that social structure, economic practices, and policies trigger climate change, while environmental protection policies are necessary. Burrows & Kinney (2016) [85] revealed that the economy and policies implemented increase climate change. IPCC (2022) [19] also draws attention to the need for effective practices to prevent climate change and migration flows in the Mediterranean Basin and North Africa Regions. Moreover, the reticence to accede to international environmental conventions also indicates that these practices need to be implemented in earnest. Some states prefer not to sign conventions (Kyoto Protocol, Climate Change Agreement, etc.),

even if they have the highest per capita emission values, while others do not fulfill the requirements of the conventions they have signed. In this context, the research draws attention to an important gap in this field.

According to the research results, when climate refugees are compared with other refugees, it is seen that a more moderate approach is taken to climate refugees. Black & Collyer (2014) [86] emphasize that people turning to limited spaces to escape danger increases their vulnerability. The fact that climate refugees have limited opportunities to escape from danger in their current regions may have led them to be seen as more vulnerable. Bansak et al. (2016) [87], in a study conducted with 18 thousand people in the European Union, found that immigrants and Christians who can contribute to the economy are preferred more than Muslims. This finding is indirectly consistent with teachers' views; Europeans are more inclined to accept religiously closer groups. Lujala et al. (2020) [88], in their study conducted in Satkhira district of Bangladesh, stated that as social distance increases, negative attitudes towards climate refugees also increase. In this study, on the other hand, teachers make positive evaluations based on the potential of climate refugees to cause less social problems compared to other refugees. In addition, teachers display positive attitudes by focusing on the ways and reasons of climate refugees' arrival. On the other hand, in the study conducted by Raimi et al. (2024) [89] in the USA, it was observed that participants who listened to the personal stories of climate refugees were limited in developing a positive perspective. This difference can be explained by the fact that the sample group in our study was able to evaluate the cause-effect relationships related to climate and migration in more detail and had a high level of empathy.

When analyzed in the context of citizenship, some of the participants abstained, stating that granting legal rights to climate refugees could be abused. Another group argues that these individuals should be granted citizenship, emphasizing that they are mostly victims of collective irresponsible behavior. These findings are like those of the United Nations Human Rights Committee [90,91] in the Ioane Teitiota case. In the case, Ioane Teitiota sought asylum in New Zealand as a climate refugee, but her application was rejected on the grounds of the 1951 Geneva Convention. Teitiota applied to the UN and requested citizenship on the grounds of security of life, and the UN stated that climate refugees may be entitled to asylum, but Teitiota did not meet these conditions. This decision is a case law in terms of determining the legal status of climate refugees. The two main approaches that emerged in the research findings also overlap with this case: While asylum for climate-related reasons may be possible, there is also a risk of abuse. Teachers show that although they do not have much knowledge about the concept, they are able to interpret the situation with sociological and psychological evaluations. In addition, one teacher stated that citizenship should be determined by the people through a referendum and addressed the issue with its social dimension.

In the study, teachers stated that the concept of climate refugeeism was mostly addressed indirectly rather than directly in the social studies course. While one teacher stated that he/she covered this issue directly, another teacher stated that the concept was not included. When the Social Studies Curriculum published by the Ministry of National Education [40] is examined, it is seen that although the concept of climate refugee is not directly included, the effects of climate change and environmental degradation on human settlements and movements are emphasized. This is in line with the findings of the study.

Teachers emphasized that the concept of climate refugee is important for social awareness and climate consciousness and stated that this concept should be included in the social studies course. Social studies course, by its very nature, aims to develop active citizenship, sensitivity to social problems and environmental awareness [37,38,39]. In this context, the results of the study are consistent with the function of the social studies course. Karakuş (2018) [92] states that the mission of social studies to raise effective citizens is combined with environmental education, and Özdemir (2022) states that this course is effective in developing positive attitudes towards the environment. Şeker (2024) [93] also emphasizes that social studies course is related to environmental education by covering social and global citizenship issues. In a meta- analysis study conducted by Yun-Wen Chan (2025) [36], it was found that social studies teachers were successful in adapting to civic and environmental literacy, but had difficulty in ecocentric thinking. This suggests that environment and climate issues should be taught more effectively in social studies teaching.

One key finding of the study is that social studies teachers emphasized the use of active learning techniques in teaching climate refugeeism—a climate-based and increasingly relevant cause of displacement, distinct from traditional factors such as war or ethnicity. Environmental education serves as a vital medium to address this concept. Özdemir (2022) highlights the importance of active learning in environmental education, a view supported by studies at both secondary and pre-service teacher levels [94,95,96,97,98,99]. Participants suggested that environmental trips and real-life experiences would enhance learning, aligning with Bursa's (2022) [100] findings on environmental justice. Research also shows that collaborative, activity-based approaches foster ecological awareness [101]. The significance of partnerships with universities, public institutions, and NGOs in this context was underscored [102]. Participants' support for NGO collaboration echoes Durmuş & Kınacı's (2021) [103] findings on NGOs as active learning tools. Additionally, incorporating the real-life experiences of climate refugees into the curriculum with an empathetic approach was recommended, as a means to foster empathy and promote social cohesion.

## Suggestions

1. In teacher education, topics such as climate refugeeism, environmental migration, and climate justice should be integrated into undergraduate social studies programs and offered as seminars to in-service teachers.

2. Since climate refugee issues are generally taught indirectly, the concept should be explicitly included in the curriculum.

3. Unlike current approaches, incorporating real-life stories of climate refugees can enhance the effectiveness of environmental education.

4. Education should be activity-based and inclusive, with an emphasis on public service announcements.

5. Policymakers should prioritize the issue of climate refugees.

6. Researchers should work to integrate climate refugeeism into education, highlight its developmental context, and guide how to educate affected populations.

## Supporting information

**S1 File. Knowledge and experience theme.**
(PDF)

**S1 Table. The participant.**
(PDF)

## Author contributions

**Conceptualization:** leyla dönmez bayrakcı, fatih ozdemir.

**Data curation:** leyla dönmez bayrakcı, fatih ozdemir.

**Formal analysis:** leyla dönmez bayrakcı, fatih ozdemir.

**Funding acquisition:** leyla dönmez bayrakcı, fatih ozdemir.

**Investigation:** leyla dönmez bayrakcı, fatih ozdemir.

**Methodology:** leyla dönmez bayrakcı, fatih ozdemir.

**Project administration:** leyla dönmez bayrakcı, fatih ozdemir.

**Resources:** leyla dönmez bayrakcı, fatih ozdemir.

**Software:** leyla dönmez bayrakcı, fatih ozdemir.

**Supervision:** leyla dönmez bayrakcı, fatih ozdemir.

**Validation:** leyla dönmez bayrakcı, fatih ozdemir.

**Visualization:** leyla dönmez bayrakcı, fatih ozdemir.

**Writing – original draft:** leyla dönmez bayrakcı, fatih ozdemir.

**Writing – review & editing:** leyla dönmez bayrakcı, fatih ozdemir.

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
