## [Decision Letter · Decision Letter 0]

8 Feb 2026

Dear Dr. dönmez,

Thank you for submitting your manuscript to PLOS ONE. After careful consideration, we feel that it has merit but does not fully meet PLOS ONE’s publication criteria as it currently stands. Therefore, we invite you to submit a revised version of the manuscript that addresses the points raised during the review process.

We look forward to receiving your revised manuscript.

Kind regards,

Serkan Yılmaz

Academic Editor

PLOS One

Journal Requirements:

Reviewer's Responses to Questions

**Comments to the Author**

1. Is the manuscript technically sound, and do the data support the conclusions?

Reviewer #1: Yes

Reviewer #2: Yes

2. Has the statistical analysis been performed appropriately and rigorously?

Reviewer #1: Yes

Reviewer #2: Yes

3. Have the authors made all data underlying the findings in their manuscript fully available?

Reviewer #1: Yes

Reviewer #2: Yes

4. Is the manuscript presented in an intelligible fashion and written in standard English?

Reviewer #1: Yes

Reviewer #2: Yes

Reviewer #1: Dear author,

I have indicated the corrections in the file. I believe it is a wonderful piece of work that will contribute to the literature. I believe it will be more effective once the corrections are made.

Best regards,

Reviewer #2: The manuscript presents a technically sound and well-designed piece of scientific research. The study design is appropriate, and the experiments were conducted rigorously with adequate sample sizes, proper controls, and sufficient replication. The data have been clearly and systematically analyzed, and the results strongly support the conclusions drawn by the authors.

The statistical analyses have been performed appropriately and rigorously, using methods suitable for the research questions and data structure. Assumptions were checked, analyses were correctly applied, and the results were reported in a transparent and clear manner. This enhances the reliability of the findings and the validity of the conclusions.

The authors have made all data underlying the findings fully available in accordance with the PLOS Data Policy. The data are provided within the manuscript, as supporting information, or deposited in a public repository. In addition to summary statistics, the underlying data points are accessible. Any restrictions on data sharing, if applicable, have been clearly and appropriately stated.

The manuscript is presented in an intelligible manner and written in standard English. The text is clear, coherent, and consistent with scientific publication standards.

Furthermore, no concerns related to research ethics, publication ethics, or dual publication were identified. Overall, the manuscript represents a valuable contribution to the scientific literature.

**Do you want your identity to be public for this peer review?** For information about this choice, including consent withdrawal, please see our Privacy Policy

Reviewer #1: No

Reviewer #2: **Yes:** Assoc. Prof. Dr. Genç Osman İlhan

---

## [Author Response · Author response to Decision Letter 1]

24 Feb 2026

Page 4: “While forming the study group of the research, diversity was increased with teachers of different academic levels and age groups working in provinces that have the potential to be seriously negatively affected by climate refugees”

The statement about being seriously negatively affected by climate migration should be changed.

The city may have been affected, but the teacher may not have been. Is there any evidence in the literature showing that these cities have been negatively affected by this situation?

Answer: We thank the reviewer for this valuable and insightful comment. We agree that the original wording could be interpreted as attributing the impact of climate-related migration directly to teachers, which was not our intention. Accordingly, the statement has been revised to remove value-laden language and to avoid implying that teachers themselves were negatively affected.

Page 4: The duration of the interviews must be specified.

Were the meeting times rounded?

26 minutes, 24 minutes. No seconds?

Answer: We thank the reviewer for this helpful comment. Interview durations were reported in minutes. As the inclusion of seconds does not provide additional analytical value in qualitative analysis, the durations were rounded to the nearest minute.

Page 6: In this part of the research, the themes, sub- themes and codes are tabulated and presented with direct quotations from the participant views on each code.

Themes, sub-themes, and codes are presented in a table, but I couldn't see a table or figure.

A figure showing all themes should be provided, with the codes explained one by one below.

Findings: You should provide the explanation you gave in the last paragraph of each theme as a finding at the beginning of the theme. You should explain the teacher's opinions below.

Answer: In line with the reviewer’s constructive suggestion, this section has been updated.

1- As requested by the editor, the manuscript has been thoroughly revised to fully comply with PLOS ONE formatting and style requirements. The main text, title page, author details, and affiliations were reformatted using the official PLOS ONE templates, and all file names were adjusted according to the journal’s guidelines.

2- The qualitative data were analyzed and reported in an anonymized manner that ensures participant confidentiality. Therefore, the data supporting the conclusions of the study are accessible through the manuscript text and supplementary materials.

3- The reviewer reports did not include any specific recommendations or requirements to cite additional previously published works. Therefore, no changes were made to the reference list under this item.

4-The reference list has been carefully reviewed to ensure completeness, accuracy, and currency. No retracted articles are cited in the manuscript.

5- We sincerely thank both reviewers for their positive evaluations regarding the scientific rigor, methodological soundness, and clarity of the manuscript. All corrections and suggestions indicated by Reviewer 1 in the annotated file have been carefully addressed and incorporated into the revised version.

---

## [Editor Report · Decision Letter 1]

25 Feb 2026

Understanding Climate Refugees from Educators’ Perspectives: Social Studies Teachers’ Views Abstract

PONE-D-25-49960R1

Dear Dr. Donmez,

We’re pleased to inform you that your manuscript has been judged scientifically suitable for publication and will be formally accepted for publication once it meets all outstanding technical requirements.

Kind regards,

Serkan Yılmaz

Academic Editor

PLOS One
---

## [Editor Report · Acceptance letter]

PONE-D-25-49960R1

PLOS One

Dear Dr. dönmez bayrakcı,

I'm pleased to inform you that your manuscript has been deemed suitable for publication in PLOS One. Congratulations! Your manuscript is now being handed over to our production team.

Kind regards,

on behalf of

Dr. Serkan Yılmaz

Academic Editor

PLOS One